# Using Front-Face Fluorescence Spectroscopy and Biochemical Analysis of Honey to Assess a Marker for the Level of *Varroa destructor* Infestation of Honey Bee (*Apis mellifera*) Colonies

**DOI:** 10.3390/foods12030629

**Published:** 2023-02-02

**Authors:** Mira Stanković, Miloš Prokopijević, Branko Šikoparija, Nebojša Nedić, Filip Andrić, Natalija Polović, Maja Natić, Ksenija Radotić

**Affiliations:** 1Institute for Multidisciplinary Research, University of Belgrade, 11030 Belgrade, Serbia; 2BioSense Institute—Research Institute for Information Technologies in Biosystems, University of Novi Sad, 21101 Novi Sad, Serbia; 3Faculty of Agriculture, University of Belgrade, 11080 Belgrade, Serbia; 4Faculty of Chemistry, University of Belgrade, 11158 Belgrade, Serbia

**Keywords:** honey, *Apis mellifera*, *Varroa destructor*, biochemical analysis, fluorescence spectroscopy

## Abstract

*Varroa destructor* is a parasitic mite responsible for the loss of honey bee (*Apis mellifera*) colonies. This study aimed to find a promising marker in honey for the bee colony infestation level using fluorescence spectroscopy and biochemical analyses. We examined whether the parameters of the honey samples’ fluorescence spectra and biochemical parameters, both related to proteins and phenolics, may be connected with the level of honey bee colonies’ infestation. The infestation level was highly positively correlated with the catalase activity in honey (r = 0.936). Additionally, the infestation level was positively correlated with the phenolic spectral component (r = 0.656), which was tentatively related to the phenolics in honey. No correlation was found between the diastase activity in honey and the colonies’ infestation level. The results indicate that the catalase activity in honey and the PFC1 spectral component may be reliable markers for the *V. destructor* infestation level of the colonies. The obtained data may be related to the honey yield obtained from the apiaries.

## 1. Introduction

Honey is considered to be a high-quality food; thus, honey quality and adulteration are the focus of many research programs [1]. The global loss of honey bee colonies is another essential subject related to honey bees and honey production [2]. Honey bees (*Apis mellifera*) are responsible for pollinating the widest range of crops of all pollinator species. Bees contribute significantly to the achievement of sustainable agriculture. Pollinators affect an estimated 35% of global agricultural land, supporting the production of 87% of the leading food crops worldwide. Additionally, bees help maintain biodiversity and a vibrant ecosystem. The diversity of bees and other pollinators worldwide is falling. This trend is caused by various factors, including the widespread use of unsustainable farming, mono-crop cultures, excessive use of agricultural chemicals, and the effects of climate change and variability [3]. The global loss of honey bee colonies (colony collapse disorder—CCD) occurs due to their interaction with various stressors such as pests, chemical, parasites, and pathogens [2,4,5]. *Varroa destructor* has been recognized as a potential contributor to this global crisis [2,4,5,6,7]. It was reported that there was a decrease in the production of honey with the increase in *V. destructor* in the hives [8]. There have not been extensive studies that would give an insight into the quality of honey originating from infected societies.

A screening method has recently been reported that could be useful for indirectly estimating the presence of pathogens in apiaries by detecting the presence of DNA of several bee pathogens in honey [9]. The existing methods for the determination of the *V. destructor* infestation rate based on the mite fall or on evaluating the infestation rates from adult or brood samples are only reliable for colonies with a medium to high infestation rate. The methods are not precise when the brood infestation rate is < 2% [10,11]. On the other hand, the molecular methods based on DNA extraction and analysis require specialized laboratories and a procedure for the sample’s preparation [12]. We aimed to use a combination of spectroscopic and biochemical analyses of honey samples to assess a marker for the level of *Varroa destructor* infestation of honey bee (*Apis mellifera*) colonies. Proteins and phenolic compounds are two minor but important components in honey [13,14], and they are the main emitting molecular species (fluorophores) in honey [15]. Phenolic compounds in honey originate from nectar, while proteins mainly originate from bees (2/3 of the total honey proteins), but also pollen suspended in nectar could contribute to the protein content (1/3 of the total honey proteins) [16]. Physico-chemical properties, phenolic composition, antioxidant activity, and melissopalynological data are important parameters in honey’s characterization [17]. In a previous study, it was shown that the ratio of protein and phenolic components obtained from the honey emission spectra may be a useful indicator for the level of infestation to which the honey bees were exposed [18]. Certain protein/enzyme components of honey, such as catalase, have been reported as important in honey bee immunity and also are connected to bees’ nutrition at the individual and colony level [19,20]. Thus, as a step forward, this study aimed to examine whether the parameters of the fluorescence emission spectra of honey samples, related to the proteins and phenolics in honey, the biochemical parameters (the total protein and phenolic content in honey), as well as the results of the melissopalynological analysis, may be related with the level of honey bee (*Apis mellifera*) colonies’ infestation with *V. destructor*. We also aimed to detect if the enzymes in honey that originate either from the plant source, namely catalase (from included pollen grains) [21] or from the bees, namely diastase (from bees’ salivary glands) [22], correlate with the colonies’ infestation level and/or the parameters of the fluorescence emission spectra. Based on these analyses, our intention was to combine spectroscopic and biochemical parameters in order to find a promising marker in honey for the bee colony infestation level. The spectroscopic approach is simpler than the used molecular methods and does not require sample preprocessing. In addition, we proposed that it may be sensitive enough for the detection of low infestation rates and thus more precise than the methods based on mite fall. This could be a base for the simple monitoring of colonies’ infestation degree by screening their honey samples. The emission spectra of honey samples from bee hives infested with varying degrees of *V. destructor* infestation were recorded. Chemometric methods for the extraction of pure components from the fluorescence emission spectra have been applied for the analyses of the various food samples [18,23,24,25]. Parallel factor analysis (PARAFAC) is one of the methods used for obtaining components from the fluorescence emission spectra [26]. We applied PARAFAC on fluorescence excitation–emission matrices (EEMs) of the honey samples. The total phenolic and total protein content, catalase, and diastase activity were measured in honey samples to see if and how they correlate with the bee colonies’ infestation percent and with the obtained parameters of the fluorescence spectra. Melissopalynological analysis was performed to determine the concentration of pollen grains in the honey samples and the contribution of pollen from different botanical sources.

## 2. Materials and Methods

### 2.1. Reagents and Solutions

Folin–Ciocalteu reagent, gallic acid, phosphoric acid (85%), o-diasidine, peroxidase from Horseradish Type II (150–250 U/mg), and iodine (I) were obtained from Sigma-Aldrich (St Louis, MO, USA). Sodium carbonate solution, Coomassie Brilliant Blue G-250, and ethanol (95%) were purchased from Fluka Analytical (Buchs, Switzerland). Bovine serum albumin was obtained from Biowest (Nuaillé, France). Sodium chloride and Acetate trihydrate sodium was purchased from Merck (Darmstadt, Germany). Glacial acetic acid was purchased from BetaHem, (Belgrade, Serbia). Hydrochloric acid—VWR was obtained from Chemicals (Radnor, PA, USA). Starch was purchased from CarloERBA, (Val-de-Reuil, France). Potassium iodide was purchased from Fisher Scientific (Loughborough, UK). Hydrogen peroxide was obtained from AppliChem GmbH (Darmstadt, Germany).

### 2.2. Samples

The test apiary was located at the Experimental property Radmilovac of the Faculty of Agriculture, the University of Belgrade, near Belgrade (44°75′47″ N 20°58′22″ E). The honey samples were separately collected from the ten hives at the end of May 2018. The bees collected nectar continuously throughout the season, but knowing that foraging behavior, although a collective activity, depends notably on the individual characteristics of a honey bee colony [27], such as the relation of forage preferences to the colony’s fitness and worker bees’ experience [28], it requires confirmation that variability between samples has not resulted from an individual colony preference to a particular nectar sources along the foraging season. The identification of the botanical source of each honey type was performed by the melissopalynological analysis. All honey samples were multifloral. The honey samples were stored in glass jars at room temperature in the dark before the analysis.

### 2.3. Methods

#### 2.3.1. Determination of *V. destructor* Infestation Rates

To determine the average level of infestation of adult bees with phoretic *Varroa* mite, the powder sugar shake method was applied [29,30]. The samples of adult bees were taken from each of the ten hives at the beginning of the experiment and test was repeated three times during May with an interval of seven days. The level of infestation was expressed as the number of mites in 10 g of bees, about 100 bees, using the following formula: total number of mites × 10/net weight of bees (g). The level of infestation was in the range 0–1.25 per 10 g bees.

#### 2.3.2. Melissopalynological Analysis

In order to examine the character of nectar sources for the studied bee colonies, we have performed quantitative and qualitative melissopalynological analysis. Pollen was extracted and analyzed following the harmonized methods of melissopalynology [31]. Microscopic slides were scanned until a minimum of 500 pollen grains were counted over the full width of the microscopic slide and identified using referent slides and pollen identification atlases [32,33]. The honeydew elements, i.e., algae, fungal spores and hyphae, anemophilous pollen, and the pollen of nectar-less plants [34], were also counted. The relative frequency of the identified pollen types was calculated [35]. The honeydew contribution is evaluated as the ratio between the number of honeydew elements (HDE) and the pollen of nectariferous plants (PN) according to scale provided by Louveaux [36]. A pollen concentration per 10 g of honey (PG/10 g) has been estimated from the proportion of detected pollen in approximately 2.5% of the slide surface and the known total slide surface. It is a modified Louveaux method [36] in the sense that it does not use filtering to extract the particles from the suspension.

#### 2.3.3. Determination of the Catalase Activity in Honey Samples

The catalase activity in honey samples was determined using the method proposed by Huidobro et al. [37]. The honey samples, each in triplicate, were dissolved in 0.015 M of phosphate buffer, pH 7, and dialyzed for 22 h at 4 °C against the same phosphate buffer. The catalase activity was determined based on the rate of disappearance of the substrate H_2_O_2_, which was measured spectrophotometrically using a Shimadzu UV-160 spectrophotometer (Kyoto, Japan), in the system containing o-dianisidine and peroxidase in the phosphate buffer, pH 6.1, where the aliquot (200 µL) of the dialyzed sample was added. The reaction was stopped by adding HCl and the absorbance was recorded at 400 nm in 1 cm cuvettes at room temperature. the Catalase activity was expressed in U/mg proteins. The presented values are the averages from the three repeated measurements for each sample.

#### 2.3.4. Determination of Diastase Activity in Honey Samples

The diastase activity in honey was determined spectrophotometrically at 660 nm on a GBC UV-Visible Cintra 6 Spectrometer (Dandenong, Australia), Part Number: 01-0940-00, by the Schade [38] method as proposed by the International Honey Commission [39], and was expressed as a number diastase (DN) in shade units.

#### 2.3.5. Determination of Total Phenolic Content (TotPhC)

The samples were prepared according to the slightly modified method proposed by Gašić [40]. Each honey sample (5 g) was mixed with 10 mL of distilled water at room temperature and transferred to a 50 mL volumetric flask and filled to the mark with ultrapure water. The total phenolic content was spectrophotometrically determined by the Folin–Ciocalteu method with some modifications [41]. Briefly, 0.3 mL of the sample solution and 6 mL of deionized water were mixed with 0.5 mL of Folin-Ciocalteu reagent and incubated for 6 min at room temperature. After 3 mL of 20% sodium carbonate solution was added, the sample was kept at 40 °C for 30 min before the absorbance was measured at 765 nm using a Shimadzu UV-160 spectrophotometer (Kyoto, Japan). Gallic acid was used as the standard, and the calibration curve of gallic acid was prepared in the concentration range between 0 and 250 mg L^−1^. A mixture of water and Folin–Ciocalteu reagent was used as the blank. The results are expressed as the gallic acid equivalent (GAE) per kg of honey.

#### 2.3.6. Determination of Total Protein Content (TotPrC)

The total protein content was determined using the Bradford procedure [42]. The honey samples (5 g) were diluted with distilled water (10 mL). In 5 μL of the honey solution, 200 μL of Coomassie Brilliant Blue was added. The Coomassie Brilliant Blue forms a protein–dye complex. After 5 min of incubation, the absorbance was measured at 595 nm against an albumin standard solution of bovine serum (10–100 μg/0.1 mL). The total protein content was quantified and expressed as g/kg of honey.

#### 2.3.7. Fluorescence Spectroscopy

The fluorescence spectra of the honey samples were recorded using an Fl3-221 P spectrofluorometer (Jobin Yvon, Horiba, Palaiseau, France), equipped with a 450 W Xe lamp and a photomultiplier tube. The sample was placed in a solid sample holder in the front-face configuration. The illumination’s incident angle was set to 22.5°, to minimize light reflections, scattered radiation, and depolarization phenomena. The Rayleigh masking was applied in order to reduce Rayleigh scattering from the solid sample, which limits the sensitivity and accuracy of the measurement [43,44]. The fluorescence emission spectra in the range from 280 to 550 nm were recorded at the excitation wavelengths range from 270 to 370 nm, thus obtaining the EEM which was used in further statistical analyses. The integration time was 0.1 s and the excitation and emission increment were 5 nm and 1 nm, respectively. A spectral band width of 2 nm was employed for both the excitation and emission slits.

### 2.4. Data Analysis

#### 2.4.1. Data Analysis and Modelling of Spectral Features

Fluorescence data were obtained in a form of the two-dimensional EEM for each honey sample, which are further packed into a three-way array with the following organization of modes: M1—honey samples, M2—emission spectral attributes, and M3—excitation spectral features. Parallel factor analysis (PARAFAC) was performed by the PLS Toolbox v. 7.0.3. (Eigenvector research Inc., Manson, WA, USA). The EEMs were preprocessed by removing Rayleigh scattering and other spectral artefacts. Similarly, the values of the emission intensities at the wavelengths below the excitation were removed. The empty cells in data matrices were replaced with missing data (non-assigned values) instead of zero values since this can lead to distorted PARAFAC solutions according to Bro [45].

#### 2.4.2. PARAFAC

The preprocessed EEMs of the honey samples were packed in three-way data arrays and further used for PARAFAC analysis. A detailed description of the PARAFAC is provided by Bro [23,45] and here it will be explained in short. PARAFAC is able to decompose three-way data arrays into trilinear components, which number depends on the number of fluorophores in the samples. Decomposition is basically carried out following the algorithm which minimizes the sum of the squares of residuals using the least-squares approach according to Equation (1),
(1)$$\;x_{ijk}=\overset{m}{\underset{h=1}{\sum }}a_{ih}b_{jh}c_{kh}+e_{ijk}$$
where $x_{ijk}$ represents an element of a three-way EEMs array, i.e., the measured fluorescence of a sample *i*, at the emission wavelength *j*, and the excitation wavelength *k*. Every element is decomposed to scores ($a_{ih}$) and the emission and excitation loadings $b_{jh}$ and $c_{kh}$, respectively, for each of the *m* PARAFAC components (where *h* is a number of PARAFAC components and ranges from 1 to *m*). The components should ideally describe the fluorophores in the sample, while the scores should represent their relative concentrations (abundancies). The residuals $e_{ijk}$ are the values that are not captured by the model with the selected number of components. However, the number of components and their profiles are not a priori known. Therefore, the PARAFAC model requires validation. There are two ways for the model to be validated and for the selection of a proper number of PARAFAC components. The first approach is based on the fact that the PARAFAC always yields a unique solution. This feature enables the PARAFAC model to be validated by split-half analysis [23,24,45]. This means that, if the proper model with a correct number of components was obtained by applying analysis to the primary data, the two halves of the equally split datasets should lead independently to the two PARAFAC solutions with the same number of components and similar or the same emission and excitation loading profiles as the original model. The second approach is based on parameters such as the core consistency and the percent of explained variance as the measures of the model’s quality. The core consistency represents a measure of how good the data were fitted in the PARAFAC model. It represents the similarity between two matrix factorization (core) vectors, one before model fitting and the other one after fitting. If the model fit well or the number of components is suitable, the two vectors are similar. For a full PARAFAC model, the core consistency is 100%. If the number of components is too high or too low, then the similarity between the vectors is low. Therefore, the optimal number of components is determined based on the sudden decrease in the core consistency with the increase in the number of components [23]. In this work, both validation approaches were used.

#### 2.4.3. Correlation Analysis

Correlation analysis was performed using a basic data analysis add-in for Microsoft Office Excel 365 by calculating Pearson’s correlation coefficient between the PARAFAC scores and the rest of the studied variables. Statistical significance of the correlations was estimated by Student’s *t*-test.

## 3. Results

### 3.1. Emission Spectra and PARAFAC

Figure 1 shows, as an example, the series of emission spectra of a honey sample obtained for the various excitation wavelengths. The EEMs were further decomposed by PARAFAC models which were built by a successive increase in the number of components from 1 to 5. The investigation of changes in the core consistency with changes in the model’s complexity suggested that the spectral data are best described by three components, i.e., a sudden drop in the core consistency was observed in models with more than three components (Appendix A); although, the percent of the explained variance by the modelled data was satisfactory. The core consistency of a three-component PARAFAC model was 83% and the percent of explained variance by a unique model fit and the model fit was 79.02% and 97.32%, respectfully. The non-negativity constraints were imposed in all three modes.

The loading vectors of the three PARAFAC components obtained by the decomposition of EEMs are given in Figure 2. These two should represent the inherent pure emission (a) and excitation (b) spectra of the characteristic honey fluorophores. The emission loading vector of the first PARAFAC component (PFC1) reaches the maximum at 415 nm, while the corresponding excitation loading vector reaches the maximum at 335 nm. According to previous findings [46], this component can be hypothetically attributed to the phenolic compounds in honey. In the case of the second component (PFC2), one small intensity emission maximum (at 325 nm) and one prominent emission maxima at 460 nm have been observed. The excitation loading vector of PFC2 shows the peak intensity at 375 nm. This may be attributed to the polyphenolic compounds in honey which are emitted in this region after an excitation at 375 nm [47,48]. The third component with the excitation/emission maxima at 280/330 nm could be related to the aromatic amino acids, either in free form or in proteins [49]. The emission and excitation characteristics of the loading vectors of all three PARAFAC components are presented in a more perceivable way in a form of EEM heatmaps (Figure 3a–c). It is easy to spot the characteristic regions that correspond to the phenolic compounds (the right middle of the map, 350–500 × 280–360 nm), polyphenolic compounds (the upper right corner of the map, 400–500 × 340–380 nm), and aromatic amino acids/proteins (the lower left part of the map, 300–400 × 260–300 nm). The results of the split and half analysis demonstrate the presence of three components in both parts of the data, where the loading profiles in both the emission and excitation modes closely follow each other (Figure 4). This testifies that the spectral data are the best described by a three component PARAFAC model.

### 3.2. Correlation Analysis of Spectral, Biochemical, and Melissopalynological Data

In order to explore the possible connections of spectral features with the biochemical, chemical, and palynological properties of the honey samples, especially the degree of infestation, a simple correlation matrix, based on Pearson’s correlation coefficient, was calculated between the PARAFAC scores of all the components (PFC1–PFC3), their ratios (PFC1/PFC3 and PFC3/PFC1) on one side, and the biochemical data (catalase and diastase activity, total protein (TotPrC) and total phenolic (TotPhC) content) on the other. The results are presented as the correlation table (Table 1). For the number of samples *n* = 10, the values of Pearson’s correlation coefficient greater than 0.632 and lower than −0.632 were considered to imply a statistically significant correlation between the variables at the predefined significance of *p* = 0.05. The degree of infestation is highly positively correlated with the activity of catalase (r = 0.936). The degree of infestation is positively correlated with the first PARAFAC component PFC1 (r = 0.656), originating from the phenolic compounds in honey. All three PARAFAC components are positively correlated with the catalase activity. No correlation was found between the diastase activity in honey and the colonies’ infestation level. The diastase activity exhibits a statistically significant positive correlation with PFC1 and PFC3. Additionally, PFC2 shows a significant correlation with TotPhC (r = 0.728).

Figure 5 shows the dependence of the catalase activity on the infestation degree with *V. destructor*. The results for the three samples with an infestation degree of 0.00 and the three samples with 0.054–0.058 infestations were averaged due to very close values for both the infestation and catalase activity.

There are three distinct groups of samples regarding the spectrum of the pollen recorded. The first group consists of typical unifloral honey types from the *Prunus* group of plants or *Amorpha fruticosa* (the samples with infestations of 0.00, 0.11, and 0.62). The second group consists of multifloral honey types with a notable contribution of the abovementioned nectar sources plus *Gleditsia triacanthos* (the samples with infestations of 0.00 and 0.058). The third is the sample with an infestation of 0.054%, where the pollen of anemophilous *Ambrosia* sp. dominates (Appendix A). This indicates the contamination from bee pollen gathered and stored by honeybees for feeding larvae.

## 4. Discussion

The high positive correlation of the infestation degree with the activity of catalase in honey (Table 1) implies that the level of catalase might be a promising marker for the screening of honey originating from beehives infested by *V. destructor*. The catalase activity in honey monitors changes in the infestation degree (Figure 5); for low changes in infestation levels, changes in the catalase activity are low, but at higher infestation levels, there is a higher increase in the catalase activity. The uneven variation in the infestation degree and thus the catalase activity among the colonies reflects the natural origin of the samples. Catalase in honey is of a herbal origin, mostly from the pollen grains [21,50]. It was previously observed that catalase in honey may be related to the bees’ social resistance to the diseases or some other external stressors. Peroxide is another important component of honey, having a protective function and thus being in equilibrium with catalase [19]. Several studies suggested that catalase in honey could originate from contaminant sources such as microorganisms [51]. The honey samples tested in this work were proven to be sterile (results not shown) rolling out this hypothesis. It could be stipulated that some portion of honey catalase (the enzyme being present in all living cells) could originate from bees’ external parasites such as *V. destructor* [52,53].

The positive correlation of the infestation degree with the first PARAFAC component PFC1, originating from the phenolic compounds in honey, indicates that PFC1 may be a marker for the infestation degree of beehives infested by *V. destructor*. The PARAFAC scores of all three components are also positively correlated with the catalase activity. Such a positive correlation between the catalase activity and phenolic spectral components may be due to the fact that both originate from the same herbal sources (catalase from pollen and phenols from nectar) [21,50,54]. The positive correlation between catalase and PFC3 (protein spectral component) is a confirmation that the third PARAFAC component originates mostly from proteins. Similarly, the diastase activity exhibits a statistically significant positive correlation with PFC1 and PFC3. The significant correlation of PFC2 with TotPhC is a confirmation that the second PARAFAC component originates mostly from the phenolic compounds.

The spectrum of pollen in honey samples corresponds to species flowering in spring in the study region. Generally, a pollen presence and concentration in the honey samples is not correlated with the measured parameters.

The results obtained in this study indicate possible markers for the *Varroa destructor* infestation levels of colonies. A further in-depth analysis is planned to determine more precise procedures for the application of this method.

## 5. Conclusions

This study aimed to examine whether the parameters of the honey fluorescence spectra linked to proteins and phenolics, the biochemical parameters (total protein and phenolic content in honey, catalase, and diastase activity), as well as the results of the melissopalynological analysis, may be related with the level of honey bee (*Apis mellifera*) colonies’ infestation with *V. destructor*. We also related the parameters of fluorescence emission spectra to the protein and phenolic content in honey and to the enzymes that originate either from the plant source (catalase) or from the bees (diastase). The infestation level was highly positively correlated with the catalase activity (r = 0.936, *n* = 10). Additionally, the infestation level was positively correlated with the PFC1 PARAFAC spectral component (r = 0.656, *n* = 1), which was tentatively related to the phenolics in honey. These results indicate that the catalase activity and PFC1 PARAFAC component (phenolic fluorophore) may be promising markers for *Varroa destructor* infestation levels of colonies, as a kind of biotic stress, by screening corresponding honey samples. This could be a base for the simple monitoring of colonies’ infestation degree by screening their honey samples. Our results imply that the fluorescence spectroscopy and biochemical analyses of honey samples may be used for observing the effects of the other kinds of external stressors on bee colonies.

## Figures and Tables

**Figure 1 foods-12-00629-f001:**
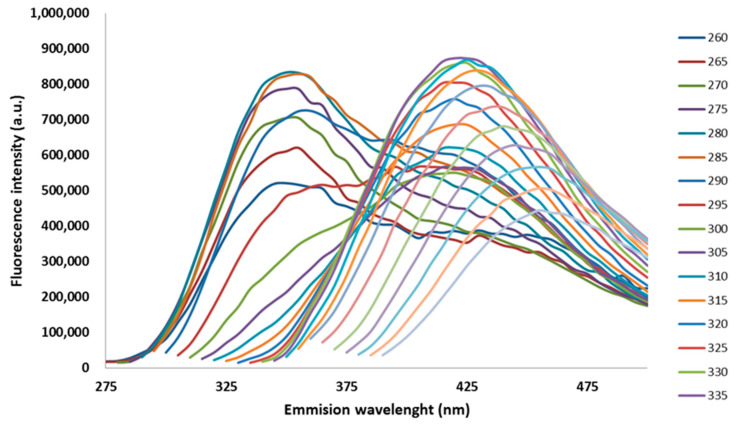
The excitation–emission matrix (EEM) for the honey sample obtained from the beehive with 0.056 *V. destructor* infestation.

**Figure 2 foods-12-00629-f002:**
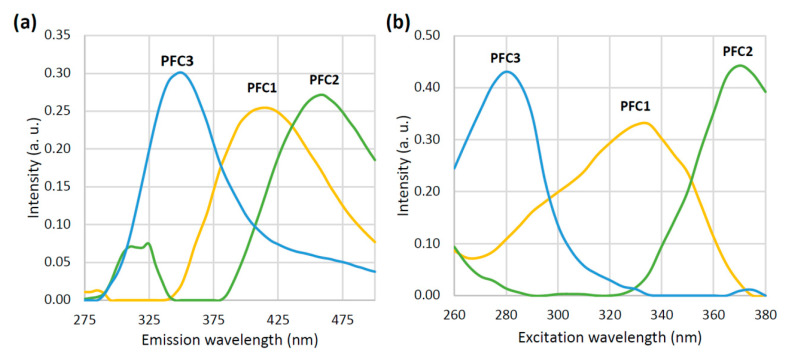
Emission (**a**) and excitation (**b**) loading vectors for the three PARAFAC components (PFC1–PFC3).

**Figure 3 foods-12-00629-f003:**
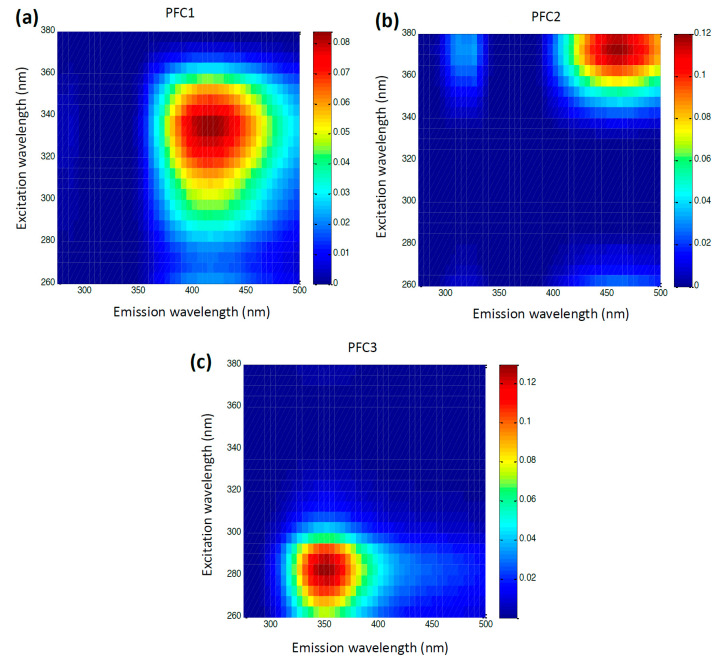
EEM heatmaps for the first (**a**) and the second (**b**) and the third (**c**) PARAFAC component corresponding to the proteins and phenolic compounds in honey samples, respectfully. Intensity scale is in arbitrary units.

**Figure 4 foods-12-00629-f004:**
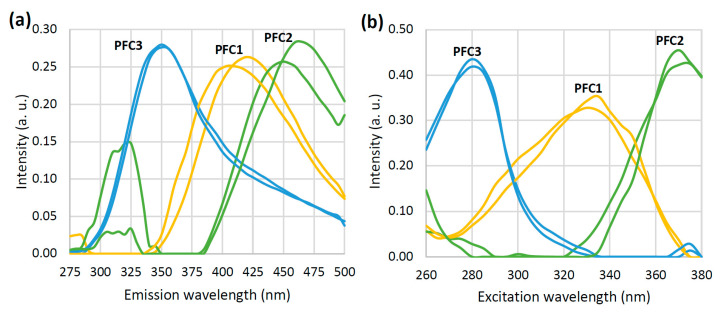
Emission (**a**) and excitation (**b**) loading vectors for the three PARAFAC components (PFC1–PFC3) obtained by the split-half analysis.

**Figure 5 foods-12-00629-f005:**
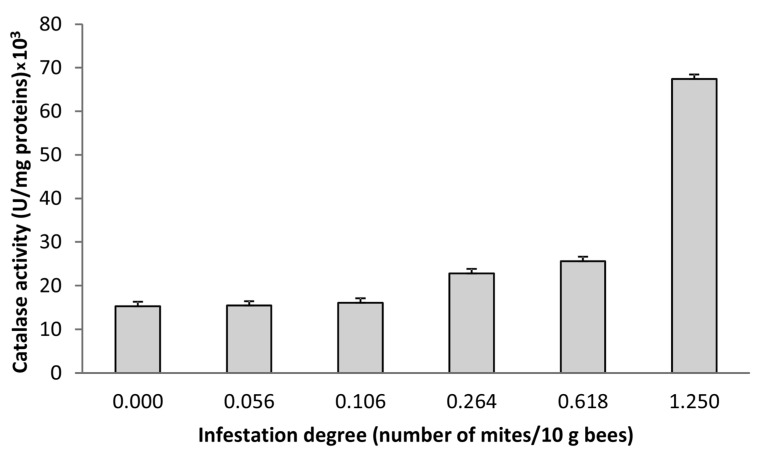
Dependence of catalase activity on the infestation degree with *V. destructor*.

**Table 1 foods-12-00629-t001:** Correlation coefficient between infestation with *V. destructor*, analyzed enzymes, proteins, phenols, and PARAFAC components (under and above the diagonal, respectively).

	Infestation	Catalase	Diastase	TotPrC	TotPhC	Pro/Phe	Phe/Pro	PFC1	PFC2	PFC3	PFC1/PFC3	PFC3/PFC1
Infestation	1.000	0.936	0.406	−0.047	−0.108	0.003	−0.036	0.656	0.513	0.608	0.061	−0.184
Catalase		1.000	0.308	0.160	0.049	0.104	−0.154	0.753	0.672	0.662	0.223	−0.272
Diastase			1.000	0.160	0.499	−0.271	0.298	0.706	0.606	0.700	−0.035	−0.080
TotPrC				1.000	0.653	0.648	−0.583	0.476	0.563	0.452	0.129	−0.066
TotPhC					1.000	−0.149	0.221	0.590	0.728	0.414	0.466	−0.425
Pro/Phe						1.000	−0.984	0.003	−0.035	0.180	−0.377	0.416
Phe/Pro							1.000	0.004	0.040	−0.181	0.376	−0.440
PFC1								1.000	0.938	0.907	0.276	−0.310
PFC2									1.000	0.753	0.521	−0.486
PFC3										1.000	−0.131	0.115
PFC1/PFC3											1.000	−0.941
PFC3/PFC1												1.000

The values of Pearson’s correlation coefficient greater than 0.632 and lower than −0.632 were considered to imply statistically significant correlation between variables at the predefined significance of *p* = 0.05. TotPrC–total protein content, TotPhC–total phenolic content, Pro/Phe–(total proteins)/(total phenols), Phe/Pro–(total phenols)/(total proteins), PFC1, PFC2, PFC3–PARAFAC components.

## Data Availability

The data are available from the corresponding author.

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
