# Peer review of "Using Front-Face Fluorescence Spectroscopy and Biochemical Analysis of Honey to Assess a Marker for the Level of Varroa destructor Infestation of Honey Bee (Apis mellifera) Colonies"

_foods, 2023, doi:10.3390/foods12030629_

Round 1

Reviewer 1 Report

This paper uses the combination of spectroscopic and biochemical analyses of honey samples to find a promising marker in honey for bee colony infestation level. It also provides an interesting topic for examining whether the parameters of fluorescence emission spectra of honey samples are related to the potential markers, such as proteins, phenolics, and biochemical parameters. However, this article will be more meaningful if it can actually realize the prediction of bee colony infection level and the identification of infection originate either from bees or plants through rapid honey fluorescence spectra ( and parameters) detection. Furthermore, this manuscript needs to be improved based on the following comments.

(1) In the introduction, please give more insight about the existing methods (such as Ref.[8]), and the comparison between the method proposed in this paper and other methods needs to be further discussed.

 (2) The description of the sample collection process is unclear. Does the only 10 honey samples collected from one covered honeycomb could be sufficient and representative for conducting the experiments? Please add one table (as supplementary materials if possible) to describe how all the sample data collected or organized in detail.

(3) It seems unconvincing by only analyzing the Pearson’ s correlation coefficients between contagion with V. destructor, analyzed enzymes, proteins, phenols and PARAFAC components to assess the marker for the level of infestation of honey bee colonies. The coefficient value of 0.632 (>0.632 or <-0.632) only indicates a certain degree of correlation, please explain why it was considered to imply statistically significant correlation between variables at the predefined significance. The experimental results and analytical conclusions only provide the possibility of revealing markers, and more reliable experimental exploration needs further in-depth analysis.

(4) Please explain what the parameter m represented for in Eq.1, does it refer to the number of components ?

 (5) Which approach was finally used for model validation and selection of a proper number of PARAFAC components ?

 (6) Abbreviation of EEMs should be clearly indicated where they first appear in the text (Line 77).  EEMs was repeated several time in the manuscript.

Line 172: excitation-emission matrix (EEM) 

Line 178-179: excitation-emission matrix (EEM) 

Line 224: excitation wavelengths (EEM).

Reviewer 2 Report

The main question addressed by the research is that whether fluorescence spectra from honey could be quantitatively correlated to infestation by V. destructor

The work have some new insights regarding the correlation of fluorescence spectra and honey infestation, keeping in mind that the correlation shown doesn't seem very strong.

Even though results are not very sound interesting, I recommend to be addressed in the abstract and conclusions.

Conclusions are consistent but I highly recommend to explicitly address the correlation coefficient value and the number of samples.

Major changes

Authors should address the numerical value of the correlation coefficient for PFC1 and the total numbers of samples analyzed in the abstract and conclusions. Also, it should be highlighted, in both sections, that the assignation of PFC1 to phenolic compounds is tentative.

Since core consistency seems to be an important parameter for building the model, a more precise explanation should be added to the manuscript.

Minor changes

Lines 37-40: repeat almost the same thing.

Line 77: Define EEMs here.

Line 104: Italics.

Line 153: superscript.

Line 167: Ok to degrees but, ¿¿Celsius??

Line 168: Rayleigh

Half of Table 1 could be deleted since values, as expected, are mirrored through the main diagonal.

The paper is well written.

Reviewer 3 Report

Dear authors,

The manuscript entitled “Using front-face fluorescence spectroscopy and biochemical analysis of honey samples to assess a marker for the level of Varroa destructor infestation of honey bee (Apis mellifera) colonies” is an interesting and well-structured study that presents a new and promising method to assess the degree of Varroa infestation in honeybee colonies through the produced honey. The catalase enzyme and phenolic compounds are shown as reliable markers of this infestation level. Also, it has an adequate literacy basis. However, it needs some minor revision to improve its understanding, following these suggestions:

-In title, the term “samples” would be deleted. It is not necessary to indicate they are samples.

- In the abstract the authors set that the obtained data may be related to honey quality and safety as food. However, no information about this topic has been exposed in the introduction or discussion. Please, add content on this subject. In other words: In which way does Varroa influence on quality and safety of beehive products?

-Line 32: Is it 87%?

-Line 37: FAO (2020) is not referenced in the list.

-Lines 39-40: This sentence is a repetition of the above. Please, delete it.

-Lines 52-53: Please check the writing of 2/3th and 1/3th. The normal expression is just 2/3 and 1/3.

-Line 75: It would be [19-22] instead of [19,20,21,22].

-Line 77: Please, develop EEMs this first time, and delete in line 172.

-Line 96:  Is it tested instead of test?

- Lines 99-101: Authors state “The identification of the botanical source of each honey type was performed by the melissopalynological analysis.” However, if all samples come from the same apiary there is one unique type. Please, explain it.

-Line 117: Is it reference instead of referent?

-Line 161: What does mean (10 100 microg/0.1 mL)?

-Lines 247-248: The third component has the emission/excitation maxima at……..Please rewrite.

-Line 252: Is it left middle of the map, or right? Please check it.

-Line 287: Please, write infestation instead of contagion.

-Line 297-304: Even if pollen analysis is not related to the Varroa infestation authors show the results. However, I do not understand different types of honeys coming from the same apiary and at the same time. Please, check the percentages of infestation, they must be the same in text, table S1 and figure 5. I did not find the specie Ambrosia dominates.

-Lines 303-304: “This indicates contamination from bee pollen gathered and stored by honeybees for feeding larvae”. I do not understand the meaning of this sentence. Please, explain it.

- In conclusions authors repeat the safety context of this study but no comment on this appears. Please add this content through the manuscript.

Round 2

Reviewer 3 Report

Dear authors,

Now, the manuscript presents all my corrected indications.